# Finding High-Quality Metal Ion-Centric Regions Across the Worldwide Protein Data Bank

**DOI:** 10.3390/molecules24173179

**Published:** 2019-09-01

**Authors:** Sen Yao, Hunter N.B. Moseley

**Affiliations:** 1Department of Molecular & Cellular Biochemistry, University of Kentucky, Lexington, KY 40536, USA; 2Markey Cancer Center, University of Kentucky, Lexington, KY 40536, USA; 3Resource Center for Stable Isotope Resolved Metabolomics, University of Kentucky, Lexington, KY 40536, USA; 4Institute for Biomedical Informatics, University of Kentucky, Lexington, KY 40536, USA; 5Center for Clinical and Translational Science, University of Kentucky, Lexington, KY 40536, USA

**Keywords:** metalloprotein, electron density analysis, pdb-eda, metal binding site, regional protein structure analysis

## Abstract

As the number of macromolecular structures in the worldwide Protein Data Bank (wwPDB) continues to grow rapidly, more attention is being paid to the quality of its data, especially for use in aggregated structural and dynamics analyses. In this study, we systematically analyzed 3.5 Å regions around all metal ions across all PDB entries with supporting electron density maps available from the PDB in Europe. All resulting metal ion-centric regions were evaluated with respect to four quality-control criteria involving electron density resolution, atom occupancy, symmetry atom exclusion, and regional electron density discrepancy. The resulting list of metal binding sites passing all four criteria possess high regional structural quality and should be beneficial to a wide variety of downstream analyses. This study demonstrates an approach for the pan-PDB evaluation of metal binding site structural quality with respect to underlying X-ray crystallographic experimental data represented in the available electron density maps of proteins. For non-crystallographers in particular, we hope to change the focus and discussion of structural quality from a global evaluation to a regional evaluation, since all structural entries in the wwPDB appear to have both regions of high and low structural quality.

## 1. Introduction

Metal ions are important components in biological processes, especially at the biochemical and cellular levels. An estimated 30% to 40% of proteins across the combined proteome of the biosphere binds at least one metal ion [1,2]. Protein metal binding is part of many biochemical mechanisms including signal transduction, enzyme catalysis, and protein structural integrity [3,4,5]. The local protein structure environment around bound metal ions can provide clues to the biochemical and cellular function of the binding [6,7,8] and reveal how sequence-based structural changes modulates metal binding [9,10]. However, the quality of 3D protein structural data around metal binding sites can vary dramatically from structure to structure, and especially from region to region [8,11]. Therefore, when analyzing metal binding site structure and dynamics, the quality of the utilized worldwide Protein Data Bank (wwPDB) [12] entries should be evaluated, especially in the metal binding site region [2,8,13]. Moreover, the presence of non-biologically bound metal ions in the wwPDB entries due to crystallization conditions and artifacts make evaluation imperative. The potential impact of crystallization artifacts on computed ligand binding affinities has already been demonstrated [14]. Also, current methods and tools for this regional evaluation around metal ions, focused only on the PDB structural entry itself, have proven useful for weeding out some metal binding sites with poor regional structural quality [13]. The best approach for identifying and evaluating a modeled metal ion is during model building and refinement using occupancy, B factor, anomalous scattering, and the chemical environment [15]. However, after a structure is deposited into the wwPDB, comparison to the raw electron density itself represents the best standard of evaluation against experimental data that can demonstrate the reliability and usability of a given metal binding site region [8,16], given the data most often available in the wwPDB. These comparisons of the metal binding site structure to the underlying electron density data have been facilitated by structure factor deposition requirements of the wwPDB since 2008, by the electron density maps made available previously by the Uppsala Electron Density Server [17], and now by the PDB in Europe (PDBe) [18]. Still, this electron density evaluation of regional structural quality has been a tedious process done by manual visual inspection, without objective metrics of quality. To alleviate these shortcomings in electron density evaluation, we have developed new analysis and evaluation methods in a Python package called pdb-eda [19], which facilitate the systematic quality control of protein structural regions of interest across large numbers of wwPDB entries and their corresponding electron density maps. In this study, we apply pdb-eda to a systematic electron density analysis of all metal binding sites containing a bound metal ion. This analysis provides an evaluation of the quality of metal binding sites in the wwPDB based on the metrics of regional structural quality with respect to the underlying electron density data used to derive the metal binding site structure. Our goal is to provide an approach for evaluating metal binding sites against experimental electron density data that could improve the outcomes for a wide variety of downstream structural, dynamic, and functional analyses. Potential downstream analyses that could benefit include, but are not limited to, molecular dynamics simulations [20], molecular mechanics and quantum mechanical calculations [21,22], and molecular docking [23,24]; however, any potential downstream analysis of metalloprotein structure that treats the PDB entry as ground truth would benefit from this type of experimental evaluation of a region of interest. Moreover, we demonstrate our evaluation approach with the generation of a current set of metal binding sites that are of high regional quality, also enabling others to screen this set with more stringent criteria specific to their data analysis needs or even to regenerate the set with a future version of the wwPDB.

## 2. Methods

Structural data from wwPDB listed on 3 July 2018 were used for the analysis. Their electron density data, if available, were acquired from the PDBe website. We used version 1.0 of the pdb-eda Python package [19] to analyze all downloaded PDB entries and matching electron density maps. Metal ions were detected across these PDB entries and filtered against four major quality control criteria: (1)Electron density resolution less than or equal to 2.5 Å;(2)Atom occupancy greater than or equal to 0.9;(3)No symmetry atoms within 3.5 Å;(4)The sum of discrepant electrons within a 3.5 Å region surrounding the metal ion point position is less than the data-derived cutoff.

The resolution and occupancy information were retrieved directly from the PDB structure entry file in PDB Molecular Format (ent) format. We considered a resolution less than or equal to 2.5 Å and an occupancy greater than or equal to 0.9 as meeting an acceptable level of quality for most downstream structural and dynamic studies, since water and small ligands are typically visible below this resolution. Symmetry-related atoms were calculated from the REMARK SMTRY records in the PDB structure data, as we took into account nearby asymmetric units. Atom–atom distance between a metal ion and all symmetry related atoms were computed and metal ions were filtered out if any symmetry atom point positions were present within 3.5 Å of the metal ion point position. Electron density maps were analyzed using the self-developed Python package, pdb-eda. This package provides methods for converting the electron density discrepancies in the experimentally observed minus calculated difference Fo-Fc density maps into numbers of discrepant electrons when a significant protein component exists in the entry. The sum of the absolute value of both positive and negative electron discrepancies was computed for all significant discrepancies within 3.5 Å of the metal. Significant discrepancies were decided by a standard deviation cutoff of 3, based on each individual electron density map, which is the commonly accepted cutoff for visualizing significant electron density map discrepancies. After filtering by criteria 1 and 2, we derived the distribution of all metal ion electron discrepancy sums and filtered out outliers based on a standard deviation cutoff of 2. With the resulting filtered distribution, we set a max electron discrepancy sum cutoff to 1 standard deviation of this distribution. The electron density overlay graphs were prepared using the LiteMol Viewer [25]. All results and code used to generate the results for this study are available on FigShare: https://doi.org/10.6084/m9.figshare.8044451. All code was run on a 20-core Intel(R) Xeon(R) E5-2670v2 CPU with hyperthreading and 256 GB RAM, utilizing all hyperthreaded cores. It took roughly 2 days to analyze the first three criteria, while it took roughly 14 days to analyze the electron density criterion for the entire PDB. 

## 3. Results

We started with a list of 141,616 usable PDB entries, and 53,146 of them contained at least one metal ion, including both biological and non-specific metal ions. The total was about 38%, which agrees with other studies and predictions [1,2]. In this study, we considered four major criteria in filtering “high-quality” metal ions, usable for downstream structural and dynamic analyses: resolution, occupancy, presence of symmetry atoms, and significant discrepancies in terms of numbers of electrons. Figure 1 shows both high- and low-quality examples based on these four criteria, as illustrated in an overlay of the electron density map over the structural model. 

Table 1 shows a tabulation of 56 different elemental types of metal ions observed in the wwPDB, with respect to four quality-control criteria. Zinc is present in the highest number of PDB entries, while magnesium has the highest number of metal sites. This is probably due to the presence of large numbers of magnesium ions in certain PDB entries, such as those of the ribosome [26]. Overall, nine metals had over 1000 examples across the PDB that passed the four criteria. An additional eight metals had over 100 examples that passed all four criteria. For the rest of this study, we look at each of the four criteria in more detail, with respect to five of the most important and common metal ions in biochemistry: zinc, calcium, iron, manganese, and copper.

### 3.1. X-Ray Crystallographic Resolution

The electron density resolution represents an overall metric of structural quality for an X-ray crystallographic structure. Structural entries with a resolution below 2.5 Å are generally considered usable for many structural and dynamics analyses. Figure 2 illustrates the distribution of resolution for the top five most essential metals in biology. In general, the overall and individual metal ions have similar distributions. The distribution for manganese has several spikes, which is mainly due to the over-representation of replicate values from structural entries with large numbers of manganese ions. This filter removes about 34% of all metal ions.

### 3.2. Occupancy and Symmetry-Related Atoms

The majority of metal ions have an occupancy of 1. However, there are two general cases where low occupancy occurs. In the first general case, when there is more than one conformation available during the structure determination, multiple conformations (typically two) are often kept in the data and are often marked as "ALT". Thus, different conformations will only possess the metal ions with partial occupancy. Typically for two conformations, the occupancy will be 0.5 for each conformation. For the second general case, only a fraction of the repeating unit cells in the protein crystal has the observed metal ion, and the occupancy will represent the percentage of the crystal structure with a bound metal ion. In either case, low occupancy sites can be considered low quality for aggregated analyses, since only a fraction of the experimental data supports the given model of the metal ion position. A filter of 0.9 occupancy removes about 10% of all metal ions and is consistent for most individual metal ions. Therefore, this criterion only removes a minority of metal ion sites.

Crystal contacts can pose as an artifact, affecting the binding of the metal ion, especially on the surface of a protein structure. This may represent a false binding that does not biologically exist, i.e., when the crystal packing environment is no longer available. Also, crystal contacts can affect protein–ligand binding [14]. Our study demonstrates that only about 7% of metal ion sites are filtered out by a 3.5 Å symmetry atom exclusion criterion.

### 3.3. Discrepancy between Calculated and Observed Electron Density Maps

Figure 1 demonstrates why electron density maps can be extremely useful for validating high-quality regions within protein structures. As described in the methods, we used our pdb-eda Python package to compare every metal binding site to its Fo-Fc electron density map. Figure 3 shows the distributions of the sum of absolute electron disagreement within a 3.5 Å radius of each metal ion. Overall and individual metal ion distributions are very similar, justifying the calculation of a single data-driven cutoff from the overall distribution. The final data-driven cutoff for the sum of absolute electron discrepancy is approximately 19.3 electrons, based on one standard deviation of the Figure 3F distribution with outliers removed. This is a purposely conservative quality control criterion, representing roughly two water molecules worth of electron discrepancy. However, only 24.8% of metal ion sites with usable electron density data (221,494) are filtered out by this criterion. In comparison, we also calculated a background difference density based on the average absolute significant electron discrepancy per Å^3^ over the whole density map, which was then multiplied by a 3.5 Å radius sphere volume. The resulting background discrepancy distribution across all metalloprotein structures is shown in Appendix A. The average number of electrons for this distribution is 1.9e, whereas the average for the metal ion sites distribution (Figure 3F) is 18.4e. Therefore, the regions around metal binding sites typically have a higher number of structural discrepancies. These discrepancies are due to experimental distortions around metal ions [27]. One possible explanation is heterogeneity in the metal ion oxidation state at a particular binding site across the crystal. Furthermore, these distortions are more pronounced around metal ion clusters, such as iron–sulfur clusters [27]. Thus, we defined metal ion clusters as any metal ion that has another metal ion within a distance of 3 Å [28,29], and then performed a similar analysis. The distribution of the electron discrepancy for metal ion clusters is shown in Appendix A. It demonstrates a very similar distribution to all categories in Figure 3, but with a higher average discrepancy of 29.6e. With the 19.3e maximum discrepancy criterion, 36.9% of cluster metal sites are filtered out. The higher level of electron discrepancy around metal ions and especially metal ion clusters emphasizes the importance of this study in finding high-quality metal-centric regions for potential downstream studies.

## 4. Discussion

As illustrated by previous studies, regional structural quality affects the usability of bound ligand structure, including bound metal ions, for accurately interpreting structural, dynamic, and chemical properties of ligand binding sites [14,30,31]. Moreover, from the comparison of electron discrepancy distributions in Figure 3F and Appendix A, metal binding regions clearly have a higher amount of electron discrepancy than the structural background. Therefore, steps should be taken to ensure the quality of metal binding sites for various downstream structural and dynamics analyses. As the number of structures available in PDB continues to grow dramatically every year, more attention is being paid to ensuring that only high-quality datasets are used in these studies. Toward this goal, we have developed new methods in the open source pdb-eda Python package that enable the evaluation of regional structural quality with respect to the underlying experimental data. 

In this study, we demonstrate the use of electron density maps for a systematic evaluation of the regional structural quality around all metal binding sites in the PDB with matching electron density maps provided by the PDBe. This is one of four criteria used for evaluating the structural quality of metal binding sites for the purpose of generating large high-quality datasets for downstream analyses. The maximum resolution criterion ensures a baseline quality of the overall structure. The combination of a minimum 0.9 occupancy criterion with a 3.5 Å symmetry atom exclusion criterion should remove most bound crystallographic artifact metal ions present in the structures, as they tend to be either inconsistently present across the asymmetric units and/or nonspecifically bound to the surface of the protein and near symmetry atoms. However, there could still be instances where a metal ion from the crystallization buffer is bound to the protein in a non-biological manner with high affinity that is comparable to bound metal ions that are biologically relevant. Distinguishing such cases requires much closer examination, often involving the use of biochemical assays, and is beyond the scope of this study. 

As demonstrated in Figure 3, metal ions and especially metal ion clusters (Appendix A) have higher levels of electron discrepancy due to the experimental distortions around metal ions. Therefore, the maximum electron density discrepancy criterion was derived from the metal ion electron discrepancy distribution itself (i.e., one standard deviation representing 19.3e). For commonly bound metal ions, additional criteria can be used for quality control [13], including the evaluation of bond lengths between ligating atoms (using the coordination chemistry definition of ligand) and the metal ion and coordination shell considerations [8,29,32]. However, several of these evaluations require the accurate identification of ligating atoms and are complicated by the wide variation in coordination geometries. Moreover, these additional criteria cannot be practically applied to all 56 elemental types of metals analyzed in this study, given the current examples available in the PDB. Therefore, we limited our method to four criteria that could be straightforwardly applied to all elemental types of metal ions currently present in the PDB.

A full list of metal binding sites that pass all four criteria utilized in this study can be found in Appendix A, along with the values used for criterion evaluation. In addition, all code used to generate this table is available in a FigShare repository along with all metals binding sites evaluated in this study. Therefore, metal binding sites can be re-evaluated against modified criteria that match the quality-control requirements of a given downstream analysis. Also with this code, future versions of the PDB can be analyzed in a similar manner to regenerate an updated list of metal binding sites.

In conclusion, we have demonstrated an approach for the pan-PDB evaluation of metal binding site structural quality with respect to underlying X-ray crystallographic experimental data represented in available electron density maps of proteins. Especially for non-crystallographers, we hope to change the focus and discussion of structural quality from a global evaluation to a regional evaluation, since all structural entries in the wwPDB appear to have both regions of high and low structural quality. 

## Figures and Tables

**Figure 1 molecules-24-03179-f001:**
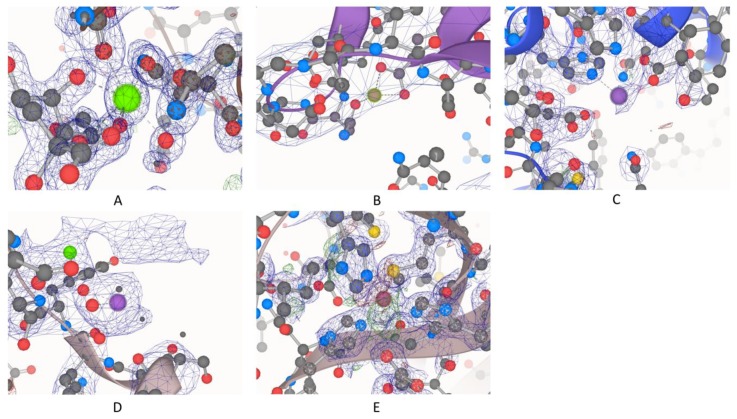
High- and low-quality examples for all four criteria: (**A**) Protein Data Bank (PDB) ID: 5FVN.F.405.CA, representative of high quality by passing all four criteria; (**B**) PDB ID: 1YV0.C.163.MG, resolution: 7 Å; (**C**) PDB ID: 3CIA.B.701.ZN, occupancy: 0.7; (**D**) PDB ID: 5ER8.A.706.MN, symmetry atoms nearby; (**E**) PDB ID: 3LZQ.A.200.CU, high discrepancy between calculated and observed electron density maps.

**Figure 2 molecules-24-03179-f002:**
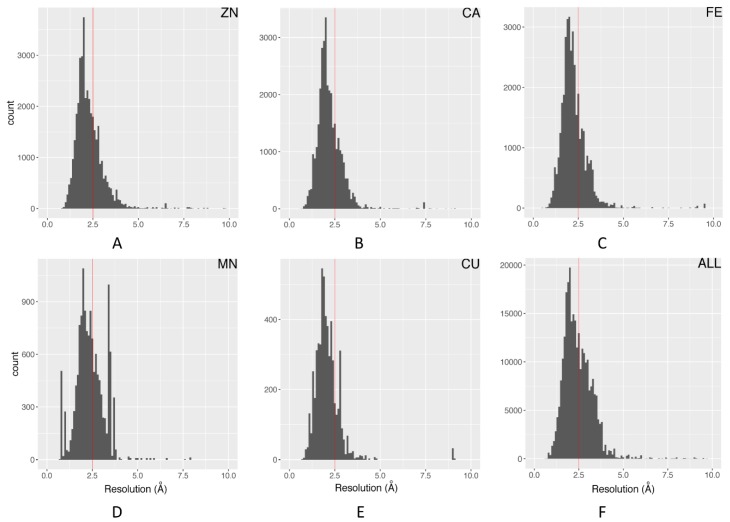
Distribution and 2.5 Å cutoff of structure resolutions: (**A**) Zn; (**B**) Ca; (**C**) Fe; (**D**) Mn; (**E**) Cu; (**F**) all metal ions.

**Figure 3 molecules-24-03179-f003:**
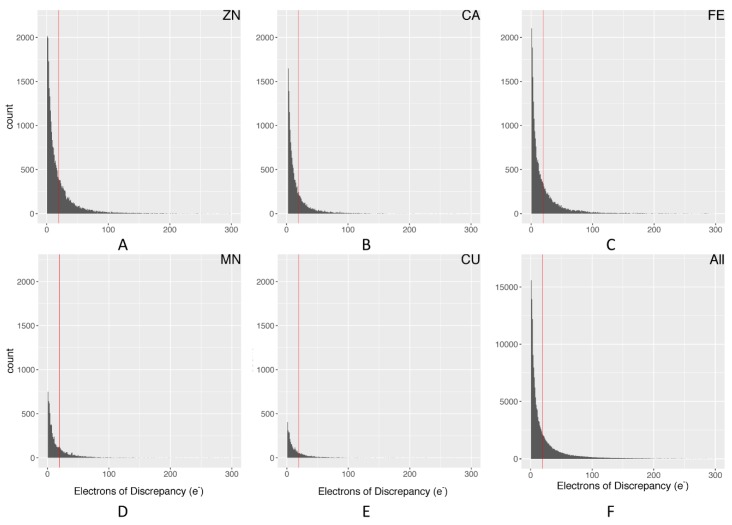
Distribution of electron discrepancy within 3.5 Å of the metal ion: (**A**) Zn; (**B**) Ca; (**C**) Fe; (**D**) Mn; (**E**) Cu; (**F**) all metal ions. The red line represents a standard deviation cutoff of 1, calculated from the distribution in graph F with outliers removed.

**Table 1 molecules-24-03179-t001:** Tabulation of different elemental types of metal ions observed in the worldwide Protein Data Bank (wwPDB) with respect to four quality-control criteria.

Metal	Number of PDB Entries	Number of Total Metal Ion Sites	Number with <2.5 Å Resolution	Number with High Occupancy	Number with No Nearby Symmetry Atoms	Number with No Significant Discrepancy Densities	Number That Passes All Criterion
Zn	13,497	67,405	56,176(83%)	62,550(93%)	62,967(93%)	23,883(35%)	13,230(20%)
Mg	13,225	85,080	30,537(36%)	81,528(96%)	81,463(96%)	48,708(57%)	18,305(22%)
Ca	10,138	44,538	36,193(81%)	41,929(94%)	41,176(92%)	23,836(54%)	15,787(35%)
Fe	8054	41,898	32,283(77%)	39,207(94%)	41,499(99%)	20,881(50%)	14,427(34%)
Na	7516	23,697	16,645(70%)	22,523(95%)	21,072(89%)	18,295(77%)	10,700(45%)
Mn	3177	14,347	9037(63%)	12,089(84%)	13,630(95%)	8755(61%)	4275(30%)
K	2390	8819	5671(64%)	7498(85%)	7678(87%)	5541(63%)	2973(34%)
Ni	1533	3578	2803(78%)	2752(77%)	2943(82%)	2093(58%)	986(28%)
Cu	1469	6918	5913(85%)	5548(80%)	6474(94%)	3676(53%)	2111(31%)
Co	1146	3601	2897(80%)	2920(81%)	3288(91%)	1687(47%)	976(27%)
Cd	926	6535	5351(82%)	4708(72%)	4863(74%)	2334(36%)	624(10%)
Hg	640	2302	1525(66%)	808(35%)	2223(97%)	528(23%)	11(0%)
U	507	6032	5522(92%)	4553(75%)	5196(86%)	2351(39%)	1693(28%)
Pt	242	869	564(65%)	212(24%)	802(92%)	249(29%)	4(0%)
Mo	209	785	685(87%)	505(64%)	692(88%)	323(41%)	147(19%)
Al	189	399	187(47%)	390(98%)	399(100%)	217(54%)	112(28%)
Be	187	510	273(54%)	461(90%)	504(99%)	318(62%)	175(34%)
Ba	166	900	399(44%)	558(62%)	733(81%)	186(21%)	6(1%)
Ru	162	341	288(84%)	163(48%)	318(93%)	113(33%)	8(2%)
Sr	151	3972	1394(35%)	3764(95%)	3869(97%)	2846(72%)	745(19%)
V	143	488	285(58%)	399(82%)	462(95%)	219(45%)	130(27%)
Cs	115	666	402(60%)	251(38%)	526(79%)	226(34%)	14(2%)
W	96	1743	396(23%)	1218(70%)	1639(94%)	280(16%)	15(1%)
Yb	91	247	189(77%)	136(55%)	127(51%)	60(24%)	6(2%)
Au	90	437	275(63%)	120(27%)	373(85%)	136(31%)	2(0%)
Li	73	124	110(89%)	116(94%)	109(88%)	96(77%)	72(58%)
Gd	65	444	408(92%)	268(60%)	409(92%)	134(30%)	22(5%)
Pb	62	229	113(49%)	87(38%)	187(82%)	63(28%)	5(2%)
Y	58	218	168(77%)	154(71%)	108(50%)	77(35%)	16(7%)
Tl	54	400	143(36%)	119(30%)	383(96%)	82(21%)	1(0%)
Ir	51	333	132(40%)	138(41%)	317(95%)	44(13%)	0(0%)
Rb	49	229	139(61%)	73(32%)	174(76%)	83(36%)	4(2%)
Sm	45	205	106(52%)	132(64%)	142(69%)	42(20%)	11(5%)
Ag	34	381	329(86%)	361(95%)	365(96%)	67(18%)	25(7%)
Pr	31	77	56(73%)	46(60%)	40(52%)	23(30%)	4(5%)
Eu	24	71	64(90%)	14(20%)	60(85%)	23(32%)	3(4%)
Pd	24	108	108(100%)	55(51%)	79(73%)	19(18%)	2(2%)
Os	23	101	34(34%)	77(76%)	97(96%)	20(20%)	3(3%)
Re	21	71	71(100%)	27(38%)	68(96%)	13(18%)	3(4%)
Rh	20	68	68(100%)	25(37%)	62(91%)	18(26%)	1(1%)
Tb	18	168	139(83%)	134(80%)	157(93%)	20(12%)	3(2%)
Ta	18	529	106(20%)	42(8%)	503(95%)	199(38%)	0(0%)
Lu	15	62	46(74%)	31(50%)	54(87%)	19(31%)	0(0%)
Ho	13	55	47(85%)	43(78%)	44(80%)	7(13%)	0(0%)
La	11	115	107(93%)	106(92%)	112(97%)	1(1%)	1(1%)
Cr	10	53	43(81%)	49(92%)	52(98%)	8(15%)	5(9%)
Ga	10	80	80(100%)	80(100%)	80(100%)	5(6%)	5(6%)
Sn	9	16	16(100%)	6(38%)	16(100%)	2(13%)	0(0%)
Sb	5	10	4(40%)	6(60%)	10(100%)	7(70%)	3(30%)
Ce	4	70	70(100%)	66(94%)	70(100%)	0(0%)	0(0%)
Er	3	18	0(0%)	17(94%)	0(0%)	1(6%)	0(0%)
Zr	3	31	28(90%)	30(97%)	0(0%)	0(0%)	0(0%)
In	2	3	1(33%)	3(100%)	0(0%)	1(33%)	0(0%)
Bi	2	2	2(100%)	0(0%)	2(100%)	0(0%)	0(0%)
Hf	2	44	44(100%)	43(98%)	0(0%)	10(23%)	9(20%)
Dy	1	26	26(100%)	0(0%)	0(0%)	18(69%)	0(0%)
Total	66,819	33,0448	218,698(66%)	299,138(91%)	308,616(93%)	168,843(51%)	87,660(27%)

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
