# Peer review of "Finding High-Quality Metal Ion-Centric Regions Across the Worldwide Protein Data Bank"

_molecules, 2019, doi:10.3390/molecules24173179_

Round 1

Reviewer 1 Report

In this manuscript, authors propose the criteria to select the high-quality structural models of metal binding sites in proteins from protein data bank. They are based on the electron density map, which is often overlooked by non-crystallographers. In addition, authors claim the importance of regional evaluation because the quality of 3D model is not uniform even in the single protein molecule. In these contexts, this manuscript is valuable and suggestive for protein scientists. However, it is not clear to what extent these selections are necessary for downstream analysis. What kinds of analyses do the authors assume? How different results will be obtained when the models which meet these criteria are used?

Current manuscript merely reports the results of selection using pdb-eda. I cannot recommend publication of this manuscript in MOLECULES without clear perspectives of this method.

Author Response

Reviewer 1:

In this manuscript, authors propose the criteria to select the high-quality structural models of metal binding sites in proteins from protein data bank. They are based on the electron density map, which is often overlooked by non-crystallographers. In addition, authors claim the importance of regional evaluation because the quality of 3D model is not uniform even in the single protein molecule. In these contexts, this manuscript is valuable and suggestive for protein scientists.

Response:

We thank the reviewer for their recognition of the value of our research.  We were trying to evaluate metal binding sites based on the experimental data and realized that there were no tools available to enable a regional evaluation of an x-ray crystallographic model against the electron density data made available by the wwPDB (technically the PDBe) in chemically meaningful metrics.

Issue 1:

However, it is not clear to what extent these selections are necessary for downstream analysis. What kinds of analyses do the authors assume? How different results will be obtained when the models which meet these criteria are used?

Current manuscript merely reports the results of selection using pdb-eda. I cannot recommend publication of this manuscript in MOLECULES without clear perspectives of this method.

Response:

It is well beyond the scope of this manuscript to evaluate how these new evaluative methods will impact all possible downstream analyses.  For this manuscript, we describe the application of these methods for the evaluation of metal bound regions in protein structures. This type of regional evaluation has a range of possible impact, depending on the type of downstream analysis being performed.  We highlighted this point using prior analyses that demonstrate the effects of nearby symmetry atoms in ligand docking analyses:

“The potential impact of crystallization artifacts on computed ligand binding affinities has already been demonstrated [14].”

However, we understand that the reviewer would like to see a better description of the context for the use of these methods.  Therefore, we have added the following statements:

“Our goal is to provide an approach for evaluating metal binding sites against experimental electron density data that could improve the outcomes for a wide variety of downstream structural, dynamic, and functional analyses.  Potential downstream analyses that could benefit include but are not limited to molecular dynamics simulations [20], molecular mechanics and quantum mechanical calculations [21,22], and molecular docking [23,24]; however, any potential downstream analysis of metalloprotein structure that treats the PDB entry as ground truth would benefit from this type of experimental evaluation of a region of interest.  Moreover, we demonstrate our evaluation approach with the generation of a current set of metal binding sites that are of high regional quality, but with the ability for other to screen this set with more stringent criteria specific to their data analysis needs or even regenerate the set with a future version of the wwPDB.”

Reviewer 2 Report

Yao and Mosely present the analysis of the PBD data based on protein structures that have heteroatoms or non CHNOPS elements. They refine all of the reported structures based on the resolution of the overall structure (<2.5 Å), and a number of local structural characteristics around the heteroatom. This includes occupancy, potential atoms related by symmetry and local electron density. Overall, the paper is well written and novel.

A few queries:

Could they provide the PDB IDs for the structures that met all of their criteria (as a supplement)?

This would be helpful for other researchers interested in particularly metal-binding proteins looking for structural motifs or binding site characteristics.

Was there an increased number of observed discrepancies between calculated and observed maps for Cu containing or redox active metals? They mention distortions around Fe-sulfur clusters, which would be consistent with this. This could be due to a mixture of oxidation states. For example Cu+1 and Cu+2 have different preferred binding geometries that result in a shift in position that may not have been modelled in the structure. Could they further break the distribution of Cu and Fe proteins based on oxidation state or binding moiety (Heme vs sulfur). Does the criteria used exclude structures with mixed oxidation states? It appears that the 0.9 occupancy would exclude these. Potentially the use of 5fo-fc would help to resolve structures that are ‘real’ but not modelled for mixed oxidation states or structures. Identifying these could be of great interest to other researchers.

It appears this is answered in Table 1. With 53% not having a significant unexplained electron density. If I understand correctly 3.5 sigma was used as the cut-off for the fo-fc map. Did they investigate the use of 5fo-fc maps for truly unassigned density or could this help to determine if the elements in the 5fo-fc map are really there and not modelled correctly due to oxidation states? This may be limited to the transition elements and heavier which are also the ones with mixed oxidation states and geometries.

Author Response

Reviewer 2:

Yao and Mosely present the analysis of the PBD data based on protein structures that have heteroatoms or non CHNOPS elements. They refine all of the reported structures based on the resolution of the overall structure (<2.5 Å), and a number of local structural characteristics around the heteroatom. This includes occupancy, potential atoms related by symmetry and local electron density. Overall, the paper is well written and novel.

Response:

We thank the reviewer for recognizing the novelty of the research and the quality of the written manuscript.  We developed these new regional evaluation methods after we looked for these types of methods and saw that they did not exist.

Issue 1:

A few queries:

Could they provide the PDB IDs for the structures that met all of their criteria (as a supplement)?

This would be helpful for other researchers interested in particularly metal-binding proteins looking for structural motifs or binding site characteristics.

Response:

All of the results and the code used to generate the results are available on the following FigShare repository: https://doi.org/10.6084/m9.figshare.8044451 .  This link is provided in the manuscript.  Also, the supplemental material has a Table S1 with the results that the reviewer is asking for.  However, not all of the supplemental material may have made it through the manuscript submission system in the first submission.  We have checked this on the submission of this revision and hope it makes it through this time.

Issue 2:

Was there an increased number of observed discrepancies between calculated and observed maps for Cu containing or redox active metals? They mention distortions around Fe-sulfur clusters, which would be consistent with this. This could be due to a mixture of oxidation states. For example Cu+1 and Cu+2 have different preferred binding geometries that result in a shift in position that may not have been modelled in the structure. Could they further break the distribution of Cu and Fe proteins based on oxidation state or binding moiety (Heme vs sulfur). Does the criteria used exclude structures with mixed oxidation states? It appears that the 0.9 occupancy would exclude these. Potentially the use of 5fo-fc would help to resolve structures that are ‘real’ but not modelled for mixed oxidation states or structures. Identifying these could be of great interest to other researchers.

It appears this is answered in Table 1. With 53% not having a significant unexplained electron density.

Response:

We agree with the reviewer that heterogeneity in the oxidation state of the bound metal could lead to electron density vs model discrepancies, when a single conformation model is used.  We thank the reviewer for brining this explanation up and we have added it to the manuscript”

“One possible explanation is heterogeneity in the metal ion oxidation state at a particular binding site across the crystal.”

However, we cannot directly detect the presence of mixed oxidation states at a given metal binding site from the PDB entry information, unless the author of the entry puts this information into the entry (which they most often do not).  However, again we agree with the reviewer that the 0.9 occupancy filter should exclude some bound metal regions with more than one significant binding conformation present in the crystallographic data.  We specifically picked evaluation criteria that were more objective and less dependent on PDB entry author interpretation. 

Also, one of the directions for this research is to develop methods that could detect incorrectly modeled metal ions.  We would view the detection of multiple oxidation states to be a similar, but potentially harder problem to solve.  However, both of these applications are beyond the scope of the current manuscript, which demonstrates the core ability to evaluate bound metal regions against the electron density maps in chemically interpretable metrics.   

Issue 3:

If I understand correctly 3.5 sigma was used as the cut-off for the fo-fc map. Did they investigate the use of 5fo-fc maps for truly unassigned density or could this help to determine if the elements in the 5fo-fc map are really there and not modelled correctly due to oxidation states? This may be limited to the transition elements and heavier which are also the ones with mixed oxidation states and geometries.

Response:

We actually use a 3 standard deviation (sigma) cutoff for detecting “significant discrepancy” in the Fo-Fc map.  But we evaluated a 3.5 angstrom radius spherical region around bound metal ions. 

Again, the direction we want to take this research is to develop methods that can detect incorrectly modeled metal ions.  Evaluation of the 2Fo-Fc electron density map is a little tricky, because significant positive density can be due to a correct model or something absent from the model.  But overlay of the model with the 2Fo-Fc map can help in this interpretation, but often requires a trained eye to interpret.

Round 2

Reviewer 1 Report

In the revised manuscript, the scope and possible application of this methods are added to Introduction according to my suggestion. I recommend the acceptance of this manuscript.